# Between Privacy and Utility: Navigating Inference Risks in De-Identified Health Data

Swati Kar[1], Lokesh Chinthala[2], Akram Mohammed[2], Robert Davis[2], and Shahnewaz Karim Sakib[1]

[1]Computer Science and Engineering, The University of Tennessee at Chattanooga, USA
[2]Center for Biomedical Informatics, The University of Tennessee Health Science Center, USA

*Abstract*—Protecting healthcare data from inference attacks, where adversaries deduce sensitive information from de-identified data, is critical. This study examines the vulnerability of such datasets, focusing on Tennessee facilities serving predominantly African American populations, while also incorporating analyses based on the MIMIC-III dataset representing Massachusetts. We apply differential privacy with varying $\epsilon$ values to assess its impact on statistical integrity and predictive model accuracy. Results show a clear trade-off: lower $\epsilon$ enhances privacy but degrades performance, while higher $\epsilon$ preserves utility at the cost of increased leakage risk. These findings underscore the importance of carefully balancing privacy and utility when allocating the privacy budget in clinical prediction tasks.

*Index Terms*—Precision Health, Inference Attack, Differential Privacy, Privacy-Preserving Machine Learning, AI-Driven Health Analytics

## I. INTRODUCTION

In recent years, data science applications have seen rapid growth in domains critical to sustainable development, including drug discovery, disease surveillance, and personalized healthcare [1]. These applications often depend on the continuous collection and analysis of sensitive medical data by third-party entities such as hospitals, wearable devices, and public health authorities [2], [3]. With the increasing reliance on interconnected systems and real-time health monitoring technologies [4], concerns about patient privacy have become more pressing. As the scale and resolution of health data collection continue to expand, the potential for adversaries to extract or infer sensitive information increases correspondingly. This growing risk underscores the need for a thorough evaluation of existing privacy-preserving mechanisms and their effectiveness in protecting individuals within real-world medical data ecosystems.

Consider a scenario where a state health agency releases a de-identified dataset containing demographics and health records to support research. An adversary, using public hospital statistics and regional demographics, aligns diagnosis patterns to infer sensitive conditions in individuals from a small rural county. This highlights how de-identified data can still be vulnerable to re-identification or inference attacks when combined with auxiliary information, raising concerns about the limitations of de-identification as a standalone privacy measure.

Electronic Health Records (EHRs) plays a pivotal role in digital healthcare, improving care quality across medical applications [5]. However, their digitization and sharing raise significant privacy concerns [6], [7]. Traditional anonymization methods – k-anonymity [8], l-diversity [9], and t-closeness [10] – have been widely used but remain vulnerable when attacker possesses auxiliary knowledge [11]–[13].

To ensure formal privacy guarantees, researchers have increasingly adopted differential privacy [14], [15], which adds calibrated noise to data outputs to mask individual contributions. While stronger privacy requires more noise, this reduces data accuracy – posing challenges in domains like healthcare where precision is vital. Although studies have examined the utility trade-offs [16], [17], the effectiveness of differential privacy in high-risk, real-world settings remains an open question.

This work addresses two critical challenges in privacy-preserving data sharing for health research. First, we show that de-identified real-world datasets remain vulnerable to inference attacks that reveal sensitive geographic information. Second, we evaluate the impact of applying $\epsilon$-differential privacy on the utility of these datasets, with a particular focus on the analytical accuracy of a downstream task for which the data is intended to be used. The key contributions of this work are as follows:

1) We design and implement a scalable inference attack that targets geographic subgroups within a de-identified real-world dataset, illustrating how adversaries can extract localized and sensitive information.

2) We apply differential privacy mechanisms at varying levels of $\epsilon$ and assess their impact on the performance of a representative downstream application.

3) We quantify the trade-off between privacy and utility, demonstrating that stronger privacy guarantees frequently result in substantial reductions in task-specific accuracy, particularly in healthcare-related contexts.

The remainder of the paper is organized as follows: Section II covers preliminaries and related work; Section III details the dataset, threat model, and problem statement; Section IV presents our experiments on inference attacks and the impact of varying privacy budgets; Section V concludes with key findings and future directions for balancing privacy and utility in health data sharing.

## II. PRELIMINARIES AND RELATED WORK

This section introduces key background concepts and reviews related works that highlight the limitations of current

privacy-preserving methods, including their vulnerability to inference attacks and challenges in balancing privacy and utility.

### A. Differential Privacy and Inference-Based Privacy Attacks

Differential Privacy (DP) [15] provides formal guarantees that the inclusion or exclusion of a single record in a dataset, $\mathcal{D}$, does not significantly affect the output of a randomized algorithm $\mathcal{A}(\mathcal{D})$. Specifically, $\mathcal{A}$ satisfies $\epsilon$-DP if, for all neighboring datasets $\mathcal{D}, \mathcal{D}'$ and all measurable subsets $S$:

$$\Pr[\mathcal{A}(\mathcal{D}) \in S] \leq e^{\epsilon} \Pr[\mathcal{A}(\mathcal{D}') \in S].$$

A smaller $\epsilon$ indicates stronger privacy by making the outputs on neighboring datasets more similar, thereby limiting the ability to infer the presence of any individual record. However, this often comes at the expense of the data's effectiveness for analytical or computational tasks, *which we refer to as utility throughout this paper*, highlighting the inherent trade-off between privacy and utility.

Inference attacks involve extracting sensitive information from benign data or statistics [18]. A key example is the membership inference attack by Shokri et al. [19], which determines if a specific data point was used in model training. Their method trains *shadow models* to mimic the target model, then uses a meta-classifier to distinguish member from non-member data based on model outputs – framing the attack as a binary classification task.

### B. Charlson Comorbidity Index

The Charlson Comorbidity Index (CCI) is a clinical tool used to estimate one-year mortality risk based on a patient's comorbid conditions [20]. It has been shown to correlate with outcomes such as hospital readmission, length of stay, and mortality. Each condition $c_i \in C$ is assigned a weight $w_i$, and the total score is calculated as

$$\mathrm{CCI} = \sum_{i=1}^{n} w_i \cdot \mathbb{I}^K(c_i),$$

where $\mathbb{I}^K(c_i)$ is an indicator function equal to $1$ if the condition is present and $0$ otherwise [21]. Weights range from 1 for conditions like myocardial infarction to 6 for severe illnesses such as metastatic cancer or AIDS. A higher CCI score indicates greater comorbidity burden and increased health risks. In this study, we incorporate the CCI score as a feature for predicting patient survival and stratify individuals into four categories: healthy, mild, moderate, and severe, according to the classification criteria in [22].

### C. Kullback–Leibler Divergence

Kullback–Leibler (KL) divergence [23] is an information-theoretic measure of how one probability distribution diverges from a reference distribution. It is an asymmetric measure commonly used to assess changes in output distributions resulting from mechanisms such as differential privacy or

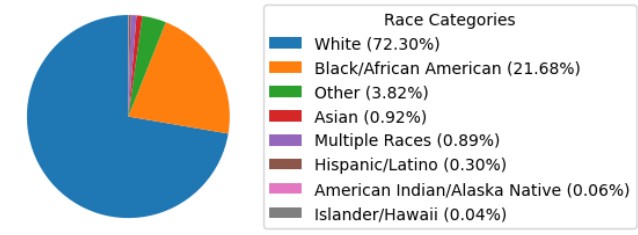

Fig. 1: Demographic distribution of breast cancer patients in the state of Tennessee. The majority of patients are White (72.3%), followed by Black/African American individuals (21.7%).

model compression. For two discrete distributions $P$ and $Q$ over the same support $\mathcal{X}$, the KL divergence is defined as:

$$D_{\mathrm{KL}}(P \parallel Q) = \sum_i p_i \log \left( \frac{p_i}{q_i} \right),$$

where $p_i = P(x_i)$ and $q_i = Q(x_i)$. The divergence is zero when the distributions are identical and infinite if $p_i > 0$ while $q_i = 0$ for any $i$.

### D. Related Works

Recent research has explored diverse strategies for securing healthcare data while balancing privacy and utility. While synthetic data generation has gained traction, studies reveal its limitations in preventing privacy breaches. For example, Chevrier et al. [24] and Stadler et al. [25] argue that synthetic data alone is insufficient, calling for integration with differential privacy (DP) and context-aware safeguards. Despite the use of privacy-preserving techniques, several works expose vulnerabilities under adversarial conditions. Mehnaz et al. [26] and Cohen et al. [27] demonstrate model inversion and re-identification risks, while Yaghini et al. [28] show that membership inference attacks disproportionately affect minority groups. El Mrini et al. [29] further reveal gradient leakage in federated learning setups, underscoring the need for stronger, demographically aware protection mechanisms.

## III. PROBLEM FORMULATION

### A. Dataset Description

We utilized seven breast cancer datasets collected from the University of Tennessee Health Science Center (UTHSC), with appropriate IRB approval, encompassing the medical histories of approximately $38,000$ patients with nearly one million records across multiple visits. Each dataset represents a distinct cohort or diagnostic context and includes:

- Anonymized patient and facility identifiers (IDs): Unique but de-identified codes used to track individuals.
- Patient demographics: Includes race, gender, and ethnicity for population-level analysis.
- Survival status: Represented by the variable `VITAL_STATUS` indicates whether the patient was alive or dead.

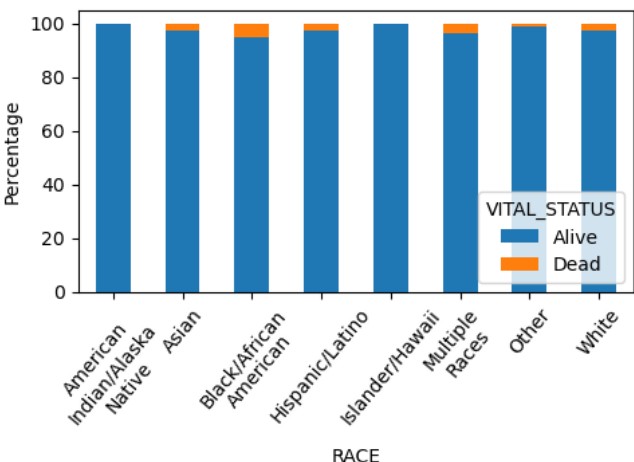

Fig. 2: Distribution of survival status (Alive vs. Dead) by race among breast cancer patients in Tennessee. The figure highlights variations in mortality rates across racial groups, with a notably higher proportion of deaths observed among Black/African American patients.

- Diagnosis codes: Admission and primary diagnoses used to assess disease patterns.
- Laboratory test results and medication records: Clinical measurements and prescribed drugs for evaluating treatment outcomes.

During preprocessing, we addressed data quality issues by handling missing values, fixing encoding inconsistencies, and removing duplicates. The datasets were then merged for unified analysis. Figure 1 shows the racial distribution of breast cancer patients in Tennessee, with White patients forming the majority, followed by Black/African American individuals. Figure 2 reveals racial disparities in survival outcomes, with notably higher mortality among African American patients, which motivated further investigation into the underlying factors contributing to these disparities.

### B. Threat Model

We are considering a threat model in which a *passive adversary* seeks to infer county-level information about breast cancer patients by leveraging de-anonymized data in conjunction with publicly available health statistics. The motivation for the adversary arises from the patterns observed in Figure 2, where disparities in survival outcomes among Black/African American patients indicate the possibility of narrowing the search space based on racial trends.

The adversary is assumed to have access only to the de-anonymized dataset introduced in III-A, which contains demographic and clinical attributes but lacks explicit geographic identifiers. *By focusing on healthcare facilities predominantly accessed by African American individuals, the attacker aims to reduce the inference space from the statewide level to a smaller set of likely counties.* To enhance inference, the adversary utilizes publicly available datasets, such as those from the National Cancer Institute [30], which

provide county-level breast cancer statistics disaggregated by race and other demographic factors. Through the integration of these public data sources and the application of statistical analysis, the adversary attempts to infer sensitive geographic details not explicitly included in the original dataset.

### C. Problem Statement

In this study, we pursue a two-fold investigation to assess the risks and limitations associated with the use of de-anonymized healthcare datasets. First, we mimic a motivated adversary who seeks to re-identify sensitive patient information. The adversary relies on patterns of facility usage and demographic disparities, with particular attention to the elevated mortality rate among African American breast cancer patients in Tennessee. By incorporating auxiliary information from publicly available sources such as the National Cancer Institute, the adversary attempts to enhance the granularity of their inferences, narrowing the search space from statewide statistics to county-level insights. This process reveals how seemingly de-identified data can still pose substantial privacy risks when combined with external knowledge.

Second, we explore the trade-off between privacy protection and data utility by applying differential privacy mechanisms to the dataset. In our evaluation, *privacy is quantified using the KL divergence between the original dataset and its differentially private counterpart obtained after applying an $\epsilon$-differential privacy mechanism. Similarly, utility is defined based on the model's ability to accurately predict patient survival status (Alive or Dead).* We analyze how prediction performance varies with privacy budgets ($\epsilon$), identifying configurations that balance privacy protection and data utility. This two-part study assesses both the feasibility of inference attacks on de-anonymized health data and the effectiveness of differential privacy in mitigating them.

### IV. EXPERIMENTAL RESULTS

### A. Inference-Based Privacy Attack on De-identified Data

We emulate an adversary aiming to narrow patient inference from state to county level by targeting facilities predominantly used by African American breast cancer patients (Figure 2), leveraging demographic disparities to improve re-identification precision. The is outlined below:

1) Identify the top ten facilities most frequently used by African American patients based on utilization counts.
2) Facilities 7752 and 7776 had the highest concentrations; facility 7752 was examined, confirming that a significant majority of patients identified as African American.
3) These facilities also reported the highest number of deaths among African American patients; reinforcing their selection, indicating a notable overlap between utilization and mortality rates.
4) Publicly available county-level cancer mortality and racial demographic data from the National Cancer Institute [30] were used to match observed trends.

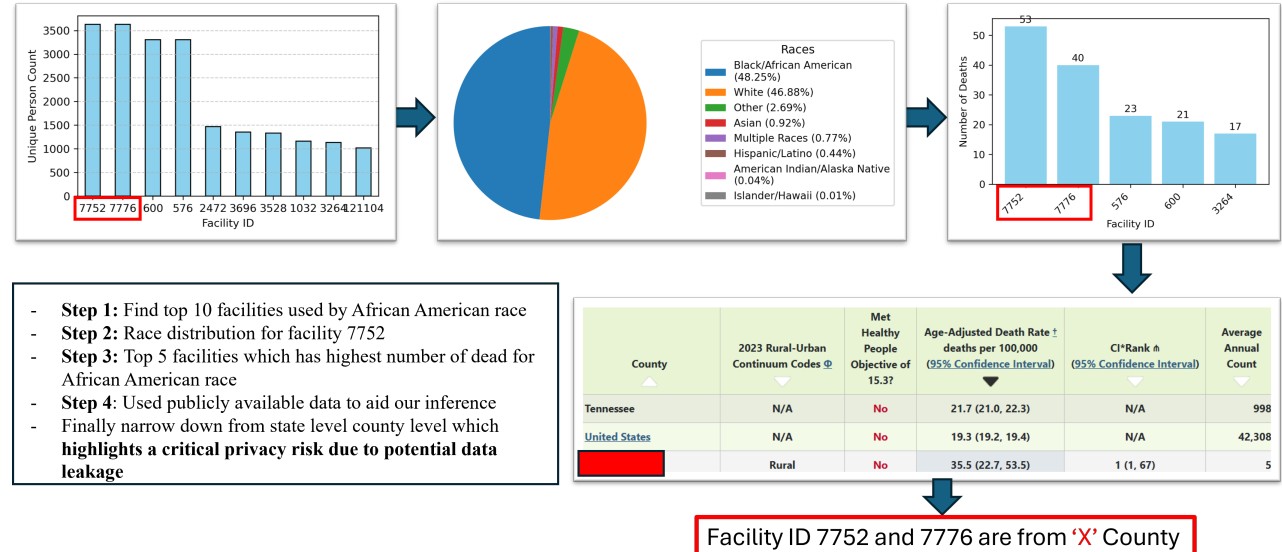

Fig. 3: Attack workflow demonstrating how the adversary (1) identifies top facilities used by Black/African American patients, (2) selects facilities with high Black/African American concentration, (3) isolates those with high reported deaths, (4) incorporates public cancer statistics, and (5) infers the likely county of the facilities. The accuracy of this inference was subsequently confirmed by the data provider UTHSC.

5) Correlating public statistics with dataset patterns enabled inference of each facility's likely county, later confirmed by UTHSC.

The full attack workflow is illustrated in Figure 3. This strategy shows a successful re-identification attempt and demonstrates the limitations of de-anonymization in preventing inference-based privacy attacks.

### B. Inference-Based Privacy Attack on MIMIC-III Dataset

To evaluate the generalizability of our inference attack, we analyzed the MIMIC-III dataset [31], which contains patient records from Beth Israel Deaconess Medical Center (Boston, MA). We assumed the data represent Massachusetts patients and validated this by comparing race and gender distributions with state demographics [32]. A Chi-square test ($\chi^2 = 0.4879$, $p = 0.9980$) confirmed no statistically significant demographic mismatch. Since MIMIC-III lacks facility identifiers, we reframed the attack question as: *Given a patient's race and disease, what is the likelihood that they reside in a specific county?* For this study, we focus on African American patients with heart disease.

To answer this question, we applied the following Bayesian inference rule:

$$P(\text{County} \mid \text{Race}, \text{Disease}) \propto P(\text{Disease} \mid \text{County}, \text{Race}) \\ \times P(\text{County} \mid \text{Race}) \quad (1)$$

Using the workflow in Figure 3, together with MIMIC-III statistics and public demographics [33], [34], we estimated the most probable counties for African American patients with heart disease. The results are summarized in Table I.

TABLE I: Top 3 counties for African American individual with Heart Disease for MIMIC-III dataset

| County | Probability |
|---|---|
| Suffolk | 0.299 |
| Middlesex | 0.161 |
| Hampden | 0.121 |

### C. Privacy-Utility Evaluation under $\epsilon$-Differential Privacy

This section analyzes how the privacy protection varies with different values of $\epsilon$, evaluates the corresponding impact on utility through predictive performance, and concludes by examining the overall trade-off between privacy and utility.

*1) Quantifying Privacy Across Varying $\epsilon$ Values:* Recall that in Subsection III-C, we quantified privacy using the KL divergence between the original dataset and its differentially private counterpart. For our experiments, we adopted four different values of $\epsilon$ – specifically, 0.01, 0.05, 0.5, and 0.9. Smaller $\epsilon$ values introduce greater noise into the data, thereby providing stronger privacy guarantees, whereas larger values reduce the amount of noise, improving data fidelity but weakening privacy protection. In this section, we focus specifically on the Tennessee facilities dataset to demonstrate the impact of differential privacy, though we expect similar effects to hold for the MIMIC-III dataset as well.

As discussed in Subsection IV-A, the adversary's objective is to localize the healthcare facilities predominantly used by African American individuals. Therefore, when quantifying privacy leakage, we restrict our focus to the subset of the dataset relevant to this demographic group. After filtering the data accordingly, we generated differentially private versions for each $\epsilon$ value. Figure 4 visualizes how patient distributions change under varying levels of differential privacy.

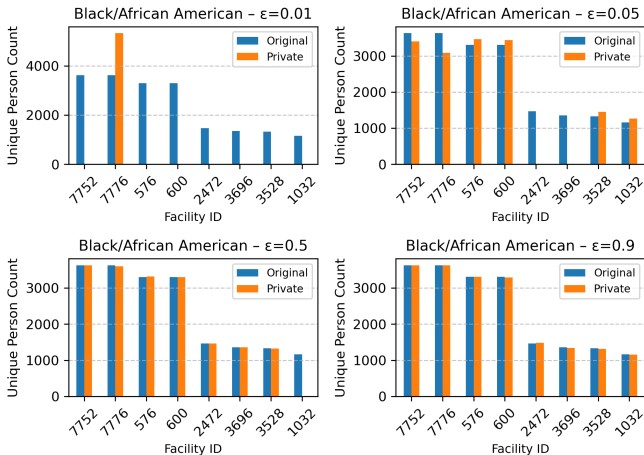

Fig. 4: Patient distributions for the Black/African American subgroup before and after applying differential privacy at $\epsilon = 0.01$ (top left), $\epsilon = 0.05$ (top right), $\epsilon = 0.5$ (bottom left), and $\epsilon = 0.9$ (bottom right). Lower $\epsilon$ values add more noise, increasing divergence from the original distribution.

A visual inspection of Figure 4 reveals that KL divergence is highest when $\epsilon = 0.01$, indicating a greater discrepancy from the original distribution, and lowest when $\epsilon = 0.9$. The corresponding KL divergence values are presented in Table III. Specifically, for $\epsilon = 0.01$, the divergence is 17.08041, and it gradually decreases with increasing $\epsilon$, reaching the lowest value of 0.000015 when $\epsilon = 0.9$. This trend highlights that lower $\epsilon$ values lead to higher KL divergence and stronger privacy guarantees, while higher $\epsilon$ values result in lower KL divergence and better data fidelity.

*2) Measuring Utility Across Varying Privacy Levels:* Now, we turn our focus to examining the impact of different privacy settings on utility. As defined in Subsection III-C, utility refers to the model's ability to correctly predict a patient's survival status *across all racial groups*. This holistic definition stems from the practical reality that the model will be deployed to predict whether a patient is alive or dead, regardless of their racial background. Consequently, we identified VITAL_STATUS as the target variable, which takes on binary values: alive or dead.

To identify the most relevant features associated with VITAL_STATUS, we computed point-biserial coefficients [35] between each feature and the binary target variable. The top eight features with the highest correlations were GENDER, ETHNICITY, RACE, FACILITY_ID, COMORBIDITY_SCORE_PRIMARY, ADMIT_DX_CD, PRIMARY_DX_CD, and COMORBIDITY_SCORE_ADMIT, while the remaining features exhibited negligible correlation. Notably, clinical variables such as COMORBIDITY_SCORE_PRIMARY and COMORBIDITY_SCORE_ADMIT, derived from diagnostic codes, play a critical role in clinical risk stratification.

Given the highly skewed nature of the dataset, where the alive-to-dead ratio is approximately $136 : 1$, we applied

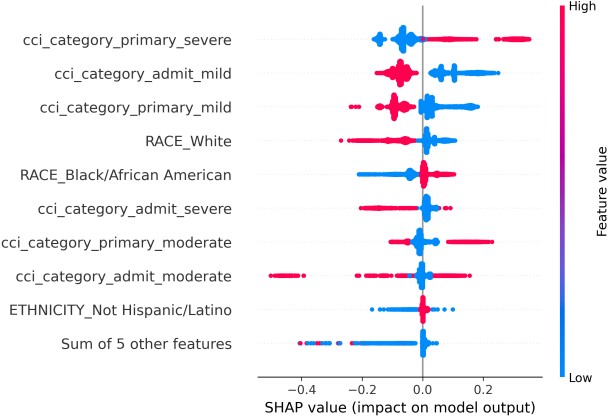

Fig. 5: Top features influencing VITAL_STATUS predictions, ranked by average absolute SHAP values.

SMOTE [36] to oversample the minority class. After balancing, we trained a neural network to assess predictive performance across different $\epsilon$ values. The model architecture is detailed in Table II.

TABLE II: Neural Network Configuration for VITAL_STATUS Classification

| Attribute | Specification |
|---|---|
| Number of Hidden Layers | 3 |
| Hidden Layer Sizes | 128, 64, 32 nodes |
| Activation Function | ReLU (for all hidden layers) |
| Output Layer Activation | Sigmoid |
| Optimizer | Adam |
| Loss Function | Binary Cross-Entropy |

To interpret how different features influence the model's predictions, we employed SHAP (SHapley Additive explanations) [37]. Figure 5 shows a SHAP beeswarm plot, which ranks features by their average absolute contribution to the model output. The most influential variables include the CCI categories, particularly cci_category_primary_severe, indicating that severe comorbidities at primary diagnosis strongly predict adverse outcomes. Racial features also show modest but distinct effects: RACE_Black/African American is linked to a higher mortality risk, whereas RACE_White slightly favors survival. The ethnicity feature ETHNICITY_Not Hispanic/Latino and other CCI categories (moderate and mild) also affect predictions, though to a lesser extent.

*3) Evaluating Privacy–Utility Trade-off:* We now evaluate the privacy-utility tradeoff across different $\epsilon$ values. As shown in Figure 6, model accuracy is about $99\%$ without differential privacy but reduces as privacy is introduced, illustrating the expected tradeoff in differentially private learning.

However, as shown in Figure 6, model accuracy remains high for larger $\epsilon$ values (e.g., 0.5 and 0.9) but drops sharply under stricter privacy ($\epsilon = 0.01$ and 0.05). Figure 7 confirms this trend: low $\epsilon$ leads to thousands of false positives due to added noise, while higher $\epsilon$ (e.g., 0.9) results in minimal misclassifications – just 6 false positives and 49 false negatives – reflecting better utility under relaxed privacy settings.

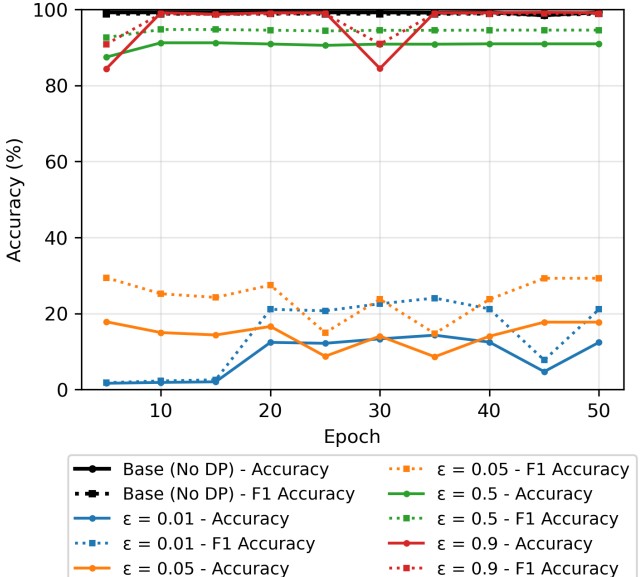

Fig. 6: Variation in accuracy and F1 score under different differential privacy settings. The figure compares model performance without privacy (no-DP) and with increasing levels of privacy ($\epsilon$), highlighting the tradeoff between utility and privacy.

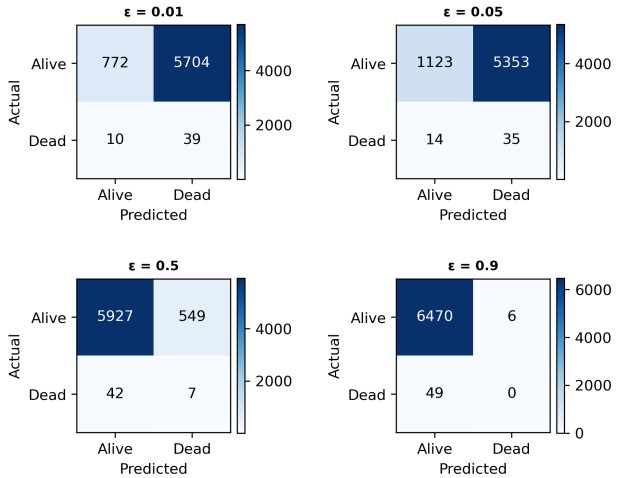

Fig. 7: Confusion matrices showing the impact of low and high $\epsilon$ values on classification performance. Lower $\epsilon$ leads to more misclassifications, while higher $\epsilon$ preserves accuracy.

Finally, Table III provides a quantitative summary of the tradeoff between privacy and utility across varying $\epsilon$ values. As expected, lower values of $\epsilon$ result in higher KL-divergence, indicating stronger privacy protection. However, this increased privacy comes at the cost of reduced predictive accuracy, reflecting a notable degradation in utility. In contrast, higher $\epsilon$ values are associated with lower KL-divergence, suggesting weaker privacy protection. Simultaneously, these settings yield higher predictive accuracy, indicating minimal utility loss. These observations are consistent with the fundamental principles of differential privacy, where

enhanced privacy often compromises model performance.

TABLE III: Impact of Differential Privacy on `VITAL_STATUS` Classification Across Varying $\epsilon$ Values, Showing the Tradeoff Between Privacy (KL-Divergence) and Utility (Accuracy).

| $\epsilon$ | KL Divergence | Test Accuracy(%) | F1 Score(%) |
|------|---------------|------------------|-------------|
| 0.01 | 17.08041 | 12.4 | 21.1 |
| 0.05 | 2.999422 | 17.7 | 29.3 |
| 0.5 | 1.223613 | 90.9 | 94.6 |
| 0.9 | 0.000015 | 99.2 | 98.8 |

### D. Guidelines for Choosing $\epsilon$ in Differential Privacy

Classical privacy-preserving techniques such as $k$-anonymity and $l$-diversity protect identity by masking or grouping quasi-identifiers. While intuitive, they often lack robustness against background knowledge and inference attacks, particularly in high-dimensional or sparse datasets [38]. In contrast, differential privacy offers a mathematically rigorous framework with provable guarantees against a broad range of adversaries. Its tunable parameter $\epsilon$ allows fine-grained control over the utility–privacy trade-off, which is critical in domains with varying sensitivity. For these reasons, we adopt differential privacy as the foundation of our study.

To provide a more concrete criterion for selecting $\epsilon$, we consider the following optimization formulation:

$$\max_{\epsilon} \quad \text{Utility}(\epsilon)$$

$$\text{subject to} \quad \tau_1 \leq D_{\text{KL}}(\text{Private info.} \parallel \text{Disclosed info.}) \leq \tau_2$$

Here, $D_{\text{KL}}$ quantifies the information leakage between private and disclosed data. The lower bound $\tau_1$ enforces that the disclosed representation is not too similar to the private information, thereby ensuring a minimum level of privacy. The upper bound $\tau_2$ ensures that the disclosed information is not overly distorted, which would otherwise degrade utility. In high-sensitivity domains (such as healthcare), $\tau_1$ may be set to a larger value to enforce stricter privacy, while $\tau_2$ can be tuned based on acceptable levels of utility degradation. For applications with lower sensitivity, this range may be relaxed to prioritize higher utility.

### V. CONCLUSION AND FUTURE WORK

In this study, we demonstrated that inference attacks leveraging publicly available demographic data can significantly narrow the geographic search space for individuals in specific subgroups, highlighting persistent privacy risks even in de-identified datasets. To mitigate this, we applied differential privacy during model training and evaluated its effect on utility in predicting patient survival. Our findings reveal a clear trade-off: stronger privacy (lower $\epsilon$) degrades model accuracy, while weaker privacy retains performance but increases vulnerability to inference threats. Future work will explore more adaptive defense strategies, including optimization frameworks that balance privacy and utility

based on user-defined preferences. We also aim to investigate context-aware noise mechanisms, adaptive privacy budgets, and adversarially informed defenses, with the goal of enhancing protection against leakage across diverse datasets and deployment scenarios.

## ACKNOWLEDGMENT

We used GPT-4o [39] to assist with rewriting and rephrasing parts of this manuscript.

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
