# OpenReview forum: "Between Privacy and Utility: Navigating Inference Risks in De-Identified Health Data"
_IEEE.org/EMBS/BHI/2025/Conference — BHI 2025_

### Official Review · Reviewer_UM9A · 2025-07-04
**Between Privacy and Utility: Navigating Inference Risks in De-Identified Health Data**

**Confidence:** 4
**Clarity Of Writing:** great
**Clinical Significance:** great
**Methodological Novelty:** good
**Overall Rating:** 7

**Experiments And Results:**

great

**Questions For The Authors:**

How would your attack and defense results generalize to other demographic groups or health conditions?
	2.	Did you consider or test other privacy-preserving techniques such as federated learning or k-anonymity for comparison?
	3.	What informed your choice of neural network architecture? Was hyperparameter tuning performed?
	4.	Can your results guide policymakers on setting ε thresholds, or do you envision this being application-specific?

**Strengths:**

Real-world relevance: Addresses pressing concerns in health data sharing and privacy, with practical implications for policy and practice.
	•	Strong experimental design: Clearly defined threat model, meaningful application of differential privacy, and thoughtful use of KL divergence and SHAP for interpretability.
	•	Detailed utility analysis: Includes SMOTE, neural network evaluation, SHAP plots, and confusion matrices to show how model performance deteriorates under strong privacy.
	•	Reproducibility elements: Includes specific architecture details and metrics (e.g., KL divergence, accuracy, F1) across privacy settings.

**Summary Of The Paper:**

This paper explores the trade-off between privacy and utility in the context of health data sharing, particularly focusing on inference risks in de-identified datasets. It presents a realistic threat model in which adversaries infer county-level information by combining released data with public statistics. The authors apply differential privacy (DP) mechanisms with varying epsilon (ε) values and evaluate how privacy affects statistical integrity and survival prediction accuracy. The paper emphasizes that while lower ε improves privacy, it significantly degrades model performance—highlighting the tension between protecting sensitive data and preserving analytical value.

**Weaknesses:**

Limited generalizability: Focuses heavily on a single subgroup (African American patients in Tennessee); results may not generalize without broader validation.
	•	Lack of baseline or comparative models: The analysis could be strengthened by comparing with alternative privacy-preserving methods (e.g., synthetic data, federated learning).
	•	Sparse discussion on policy implications: While technical results are strong, there is little commentary on how findings translate into guidelines for data custodians.
	•	No discussion on hyperparameter tuning or training stability of the neural network, which may affect reproducibility.

---

### Official Review · Reviewer_Ek5p · 2025-07-11
**Borderline ‒ De-identified TN breast-cancer data under inference attack; DP helps, but novelty and generalisability are thin**

**Confidence:** 4
**Clarity Of Writing:** good
**Clinical Significance:** fair
**Methodological Novelty:** fair
**Overall Rating:** 4
**Final Rating:** 6

**Experiments And Results:**

good

**Questions For The Authors:**

1. Would you consider rerunning the attack on a larger US dataset such as SEER or on a cohort outside the United States to show that the vulnerability is not restricted to Tennessee? A positive result would likely raise my overall score.
2. Could you benchmark other privacy defenses, for example k-anonymity with l-diversity or a synthetic-data generator, so that readers can compare them with your differential-privacy approach?
3. If you report AUROC or PR-AUC for the highly imbalanced survival task, do the privacy-utility breakpoints move?
4. Section IV-C currently provides only qualitative guidance on choosing ε. Can you turn that into a concrete rule of thumb, for instance maximising utility while keeping KL divergence below a specified threshold τ?

**Strengths:**

This study is grounded in a genuine, real-world dataset and makes a persuasive case that a seemingly modest re-identification strategy-cross-referencing public cancer statistics-can reveal which treatment facilities individual patients visited, as illustrated in Figure 3. The authors also provide a clear picture of the privacy-utility trade-off by plotting accuracy against different ε settings in Table III and Figures 6 and 7, so the reader can see precisely where predictive performance collapses as stronger privacy is enforced. A further contribution is the use of SHAP analysis in Figure 5, which helps clinicians grasp why the model’s survival predictions shift under various privacy settings.

**Summary Of The Paper:**

Using a 38 k-patient breast-cancer registry from Tennessee, the authors (i) stage a county-level inference attack that re-links supposedly de-identified records to geography, then (ii) add ε-differential-privacy noise (ε = 0.01…0.9) and track the privacy-utility trade-off on a survival-status prediction task. Stronger privacy (ε ≤ 0.05) kills accuracy; relaxed settings (ε ≥ 0.5) keep ~91–99 % accuracy but leak more. SHAP plots highlight which features drive the network’s Alive/Dead outputs.

**Weaknesses:**

Despite these merits, the scope of the work is confined to a single disease area within one U.S. state, leaving open the question of how well the findings would generalise elsewhere. The threat model assumes an attacker who already possesses detailed information on race-specific facility utilisation, a level of prior knowledge that may be unrealistic in practice. Although the manuscript mentions alternative defences such as k-anonymity or synthetic data, it does not benchmark the proposed differential-privacy approach against them, so readers cannot assess the relative trade-offs. In addition, the main utility metric—accuracy—may be misleading because the Alive-to-Dead class ratio is 136 to 1; a more informative measure would be AUROC or PR-AUC. Methodological novelty is also limited, as the work applies well-established differential-privacy noise at multiple ε levels without introducing a new mechanism. Finally, the manuscript contains a handful of typos and even an explicit acknowledgment of GPT-4o assistance, which could raise concerns for some reviewers about the writing process and overall polish.

---

### Official Review · Reviewer_KPH6 · 2025-07-17
**Review of "Between Privacy and Utility: Navigating Inference Risks in De-Identified Health Data"**

**Confidence:** 3
**Clarity Of Writing:** great
**Clinical Significance:** great
**Methodological Novelty:** good
**Overall Rating:** 5

**Experiments And Results:**

great

**Questions For The Authors:**

1. How might the results change if the analysis were extended to other diseases, populations, or geographic regions?
Given the focus on breast cancer data from Tennessee facilities serving predominantly African American patients, have you considered whether similar inference risks and privacy-utility trade-offs would emerge in less distinct or differently structured datasets?

2. What considerations guided your choice of neural network architecture and utility metrics?
Did you examine how other machine learning models or more clinically specific evaluation measures (such as AUC or calibration) might influence the observed privacy-utility trade-off, particularly with respect to clinically actionable insights?

3. How realistic is the adversary model in diverse data-sharing environments?
Could you elaborate on how widely available and granular the external auxiliary data are in practice, and have you explored scenarios in which the adversary possesses only partial or less-reliable public data?

4. Have you considered evaluating alternative privacy-preserving techniques beyond standard differential privacy?
For example, might personalized differential privacy, federated learning with DP, or synthetic data generation combined with DP offer a better privacy-utility balance—particularly for minority subgroups? If so, what preliminary observations can you share, or what challenges do you foresee?

**Strengths:**

1. Realistic and Impactful Threat Demonstration:
The paper excels by empirically demonstrating that inference attacks on de-identified health data are not just theoretical, but a viable threat in practice. By using actual clinical data and simulating a determined adversary who leverages publicly available demographic and outcome statistics, the study provides compelling, concrete evidence of the limitations of de-identification. This realistic approach underscores the urgency of the problem for policy makers, data custodians, and clinicians.

2. Comprehensive and Quantitative Privacy-Utility Assessment:
A major strength of the work lies in its systematic analysis of the trade-off between privacy and utility using differential privacy. By methodically applying varying privacy budgets (ϵ), measuring privacy loss using Kullback-Leibler divergence, and evaluating utility through clinically relevant predictive modeling (patient survival), the authors deliver nuanced insight into how privacy protections deteriorate analytic performance. This rigorous quantification, paired with practical outcome measures, significantly enhances the value of the findings for real-world healthcare analytics.

3. Clinical and Practical Relevance:
The authors ensure that their analysis and conclusions are meaningful for healthcare stakeholders by utilizing genuine clinical features (such as the Charlson Comorbidity Index) and focusing on outcomes (patient survival) that matter in medical practice. This domain grounding lifts the work above more abstract privacy studies, making the results immediately actionable for those working in precision health and medical data science.

4. Clear and Actionable Recommendations:
Beyond technical results, the paper provides thoughtful discussion and concrete guidelines about selecting privacy parameters in a healthcare context. The emphasis on aligning privacy budgets with data sensitivity and the risk of re-identification addresses the pressing need for responsible and context-aware privacy management in health data sharing policies.

**Summary Of The Paper:**

This paper examines the vulnerability of de-identified healthcare data to inference attacks, using breast cancer records from Tennessee facilities serving predominantly African American populations as a case study. The authors demonstrate that, by leveraging publicly available demographic and mortality statistics, an adversary can re-identify sensitive county-level information even without explicit geographic identifiers. Their results expose significant risks inherent in conventional de-identification, especially when external auxiliary data is available.

To counter these risks, the paper investigates the impact of applying differential privacy to the dataset at several privacy levels, quantifying privacy through Kullback-Leibler divergence and utility through the predictive accuracy of a neural network model focused on survival prediction. The study finds a clear privacy-utility trade-off: stringent privacy (low \epsilon) strongly limits leakage but substantially reduces model performance, while a more permissive privacy budget (high \epsilon) preserves analytic utility but leaves the data susceptible to inference. The use of clinical features such as the Charlson Comorbidity Index strengthens the practical relevance of the findings.

This work is notable for its realistic threat scenario and concrete demonstration of privacy risks, as well as its thoughtful discussion on the practical considerations in selecting privacy parameters for health data. However, its empirical focus on a single geography and population somewhat limits generalizability, and it evaluates only standard differential privacy approaches without exploring more advanced or hybrid mitigation strategies. The utility evaluation relies on common machine learning metrics, and the study could be further enhanced with additional clinical or operational perspectives.

In summary, this paper provides timely, empirically grounded insight into the persistent privacy risks of de-identified health data and the necessary trade-offs imposed by differential privacy. It offers clear guidance on privacy budget selection and highlights the importance of context-aware defenses, making it a meaningful contribution to the discourse on ethical data sharing in healthcare.

**Weaknesses:**

1. Limited Generalizability Due to Narrow Scope
While the paper provides a valuable case study, its empirical focus is limited to breast cancer data from facilities primarily serving African American populations in Tennessee (see Problem Formulation, Dataset Description). As a result, it remains unclear whether the demonstrated inference risks and the privacy-utility trade-offs identified would hold for other diseases, populations, or geographic regions. Expanding the evaluation to additional datasets from diverse clinical settings would strengthen the generalizability of the findings and bolster the argument that these privacy risks persist across healthcare more broadly.
The authors can consider conducting similar inference attacks and differential privacy experiments on datasets from other states and with different demographic distributions or health conditions. This would demonstrate the robustness and applicability of your recommendations to a wider audience.

2. Adversary Assumptions and Realism
The attack relies on an adversary who possesses comprehensive public health statistics and has the analytical skill to align patterns from de-identified data with external sources (see Threat Model and Experimental Results A). While plausible, this model represents a strong adversary and may not reflect the capabilities or resources of most potential attackers. The real-world likelihood of such inference may therefore be lower than presented, or may look different with less granular or less readily accessible external data.
It would be helpful to quantify or discuss more systematically the accessibility and granularity of external reference data (e.g., public records varying by state/country). Additionally, running sensitivity experiments where the adversary’s auxiliary knowledge is incomplete or noisy could shed light on scenarios where the risk is lower, thus helping data custodians better contextualize the threat.

3. The study applies standard differential privacy mechanisms (varying the \epsilon parameter) but does not experiment with more sophisticated or hybrid approaches, such as context-aware noise injection, adaptive privacy budgets, or combining DP with traditional de-identification or synthetic data (see Privacy-Utility Evaluation and Conclusion). Alternative techniques might better balance the privacy-utility trade-off or address subgroup-level vulnerabilities differently. The authors may consider implementing or at least discussing experiments with alternative privacy-preserving strategies. Investigating whether techniques such as personalized DP, federated learning with DP, or synthetic data generation combined with DP can mitigate the unique risks facing minority or small groups would be valuable. Such experiments could determine if certain approaches are better at protecting against the kinds of inference attacks demonstrated in this paper.

4. Utility is assessed primarily through neural network classification accuracy and F1 score on survival prediction (see Privacy-Utility Evaluation and Table III). While these are informative, they do not fully capture the downstream clinical relevance or the impact on patient care. Nor do they address how privacy interventions might differentially affect minority subgroups or the interpretability/robustness of predictions. Future work could incorporate additional utility measures such as AUC, calibration metrics, or clinically meaningful endpoints (for example, correct identification of high-risk patients who could benefit from intervention). Also, evaluating model performance across demographic subgroups could reveal if privacy measures disproportionately harm accuracy for already vulnerable populations, deepening the practical and ethical impact of your work.

---

### Official Review · Reviewer_7GG5 · 2025-07-20
**Review of Paper #230**

**Confidence:** 4
**Clarity Of Writing:** great
**Clinical Significance:** good
**Methodological Novelty:** great
**Overall Rating:** 7

**Experiments And Results:**

great

**Questions For The Authors:**

Please see my comments above.

**Strengths:**

Overall this paper is well-written and organized with plenty of experiments conducted.

**Summary Of The Paper:**

This paper applies differential privacy during model training and evaluates its effect on utility in predicting patient survival.

**Weaknesses:**

1) The authors state that: "Consequently, we identified VITAL_STATUS as the target variable, which takes on binary values: alive or dead.
To identify the most relevant features associated with VITAL_STATUS, we computed Pearson correlation coefficients [40] between each feature and the target."

Point biserial correlation should be used, since the outcome variable is binary.